# High-Risk Regions of African Swine Fever Infection in Mozambique

**DOI:** 10.3390/v15041010

**Published:** 2023-04-20

**Authors:** Azido Ribeiro Mataca, Francisco Alyson Silva Oliveira, Ângelo André Lampeão, José Pereira Mendonça, Maria Aparecida Scatamburlo Moreira, Rinaldo Aparecido Mota, Wagnner José Nascimento Porto, David Germano Gonçalves Schwarz, Abelardo Silva-Júnior

**Affiliations:** 1Departamento de Veterinária, Universidade Federal de Viçosa, Viçosa 36570-900, Brazil; azido.ribeiro@gmail.com (A.R.M.);; 2Escola Superior de Desenvolvimento Rural, Universidade Eduardo Mondlane, Maputo 257, Mozambique; 3Hospital Veterinário Universitário, Universidade Federal do Piauí, Bom Jesus 64900-000, Brazil; 4Instituto de Investigação Agrária de Moçambique (IIAM), Maputo 2698, Mozambique; 5Departamento de Medicina Veterinária, Universidade Federal Rural de Pernambuco (UFRPE), Recife 52171-900, Brazil; 6Centro de Ciências Agroveterinárias, Universidade do Estado de Santa Catarina (UDESC), Lages 88.520-000, Brazil; 7Instituto de Ciências Biológicas e da Saúde, Universidade Federal de Alagoas, Maceió 57072-900, Brazil

**Keywords:** geographic surveillance, animal health defense, trends, cluster

## Abstract

African swine fever (ASF) is a transboundary infectious disease that can infect wild and domestic swine and requires enhanced surveillance between countries. In Mozambique, ASF has been reported across the country, spreading between provinces, mainly through the movement of pigs and their by-products. Subsequently, pigs from bordering countries were at risk of exposure. This study evaluated the spatiotemporal distribution and temporal trends of ASF in swine in Mozambique between 2000 and 2020. During this period, 28,624 cases of ASF were reported across three regions of the country. In total, the northern, central, and southern regions presented 64.9, 17.8, and 17.3% of the total cases, respectively. When analyzing the incidence risk (IR) of ASF per 100,000 pigs, the Cabo Delgado province had the highest IR (17,301.1), followed by the Maputo province (8868.6). In the space-time analysis, three clusters were formed in each region: (i) Cluster A involved the provinces of Cabo Delgado and Nampula (north), (ii) Cluster B involved the province of Maputo and the city of Maputo (south), and (iii) Cluster C consisted of the provinces of Manica and Sofala (central) in 2006. However, when analyzing the temporal trend in the provinces, most were found to be decreasing, except for Sofala, Inhambane, and Maputo, which had a stationary trend. To the best of our knowledge, this is the first study to evaluate the spatial distribution of ASF in Mozambique. These findings will contribute to increasing official ASF control programs by identifying high-risk areas and raising awareness of the importance of controlling the borders between provinces and countries to prevent their spread to other regions of the world.

## 1. Introduction

African swine fever (ASF) is a transboundary, infectious, and highly fatal disease that affects domestic swine [1,2]. In African countries, where food shortages affect a large portion of the population, ASF has a particularly negative impact and is one of the main obstacles to swine production [3]. This disease causes severe economic losses to swine production in the small commercial sector and subsistence family farms in Africa [2,3,4]. ASF is caused by the ASF virus (ASFV) of the genus *Asfivirus*, a member of the *Asfarviridae* family, and is characterized by large virions, 175–215 nm in diameter, icosahedral symmetry, and an envelope [5,6]. The viral genome consists of a double-stranded DNA molecule, approximately 170–190 kb, depending on the strain [7], which is highly tolerant to a wide range of temperatures and pH and can survive for long periods in meat [8]. In Africa, the virus is well-adapted to its natural hosts, sylvatic suidae (*Phacochoerus aethiopicus* and *Potamochoerus porcus*) and argasids (*Ornithodorous moubata*), without causing apparent infections [7,9,10]. 

The ASF emerged in 1960 in Mozambique in the mid-west region of the country, adjacent to the border with Malawi [11], and has persisted for many years, spreading to the northern region of the country [3]. The disease spread to the southern provinces of the country in 1994 owing to the movement of people and domestic animals after peace was established in 1992 [3,10,11]. Thereafter, sporadic outbreaks have been reported more frequently in endemic areas with high densities of smallholders who have implemented few biosecurity measures related to the commercial movement of live pigs. Little epidemiological information is available on ASF in Mozambique; however, it is suspected that the virus is maintained by the sylvatic cycle in domestic swine such as wild boars and other suids [12,13]. Thus, this sylvatic cycle may occur in swine regions adjacent to the Malawi-endemic area of the Mechinje district [3].

In Mozambique, the source of infection for commercial pigs is the free movement of infected pigs from rural areas, where large numbers of these pigs are produced using conventional methods, with a risk of contagion between pigs with different properties. Although the commercial production of pigs in Mozambique is small, with more than 90% of the production belonging to poor small family farmers [3], the country is strategically located for the dissemination of ASF as it borders six other countries: South Africa, Swaziland, Zimbabwe, Zambia, Malawi, and Tanzania. The 2010 agricultural census indicated that the highest concentration of pigs was found in the central–north region of Mozambique, where commercial pig farming is infrequent [14]. Since ASF was first identified, several studies have sought to clarify different aspects of its biology and pathogenesis [1,15]. However, no effective and safe vaccine has been identified due to its low immunological induction and inability to generate proteins capable of differentiating vaccinated animals from unvaccinated animals [16,17,18]. 

Currently, disease control is based exclusively on strict prophylactic sanitary measures. Measures such as (i) compulsive slaughter of infected and contacting animals, (ii) sanitary breaks, and (iii) bans on exporting products of animal origin have high economic costs for countries affected by ASF. Implementing preventative measures for ASF and other infectious diseases has proven to be difficult, especially in the informal sector. In Africa, in addition to trading through official channels, pigs and their meat are exchanged through informal events, such as ceremonies and family visits. Pig farmers and residents in the informal sector must have sufficient knowledge of diseases and preventive measures. Therefore, health education programs in this sector should continue, particularly for community leaders and agricultural extension agents. Similar to many African countries, in Mozambique, most ASF outbreaks in domestic swine have not been reported, and many other suspected outbreaks have never been laboratory-confirmed [3], mainly in the informal sector, intensifying the underreporting of official cases. However, in Mozambique, spatiotemporal analysis allows the identification of clusters with a higher risk of infection and a longer persistence time by identifying vulnerable areas, thereby contributing to official eradication programs and the implementation of active measures to control the disease [19]. In this context, this study sought to identify the regions with the highest risk for ASF through space-time analysis and assess temporal trends in different provinces of Mozambique over 20 years (2000–2020).

## 2. Materials and Methods

### 2.1. Study Area 

The Republic of Mozambique is a country in Southern Africa located on the southeastern coast of Africa between the latitudes of 10° and 27° S and 30° and longitudes of 41° E. The country spans a total area of 799,380 km^2^, bordered by Swaziland to the south, South Africa to the southwest, Zimbabwe to the west, Zambia and Malawi to the northwest, Tanzania to the north, and the Indian Ocean to the east [20] (Figure 1). Mozambique is administratively divided into 11 provinces, 151 districts, 405 administrative posts, and 1048 rural locations. Since 1998, 53 municipalities have joined these divisions [20]. The national livestock population is concentrated in the central provinces of the country, mainly between the provinces of Tete, Manica, Sofala, and Zambezia, which have 48.97% (48.03% pigs) of the national size, representing approximately 15,019,313 heads (644,002 pigs) [14].

### 2.2. Data Sources

Data from the 11 provinces of Mozambique were provided by the National Directorate of Livestock Development (DINADP), which operates under the auspices of the Ministry of Agriculture and Rural Development (MADER). These data were collected in the field by veterinarians in each of the 151 districts of Mozambique with reported ASF cases from 2000 to 2020. Once collected, data were transferred monthly to the DINADP and monitored by the official veterinary services of each district. This study included data on the reported and confirmed cases of ASF in domestic pigs, regardless of the breed. Information, such as the place of notification (district and province), number of reported or confirmed cases, year of notification, and geographic coordinates, was collected. Each notification represented a new case. There was no overlap between the notifications. The total annual number of pigs in each of the 11 provinces during the study period was obtained from the same database. All data were organized in a Microsoft Excel 2010 spreadsheet. The crude and swine census data can be accessed in Appendix A.

### 2.3. Incidence Risk and Temporal Trend Analysis

The relative frequencies (RF) and incidence risks (IR) of ASF for each province and year from 2000 to 2020 were calculated and are presented in tables, graphs, and maps. RF was calculated by dividing the total number of cases registered in the province by the total number of cases from all provinces (countries) and multiplying by 100. IR was calculated by dividing the total number of new cases by the total number of pigs in each province and year and multiplying by 100,000 to obtain an integer. This is because the high value used as the denominator resulted in a low percentage of evaluations. Therefore, in this study, IR values were used proportionally. Data normality was assessed using the Shapiro–Wilk test, with a significance cutoff of 0.05, using the PAST software, ver. 3.25 [21]. The temporal trends in the IR of the provinces for the years of the study were analyzed using the Joinpoint Regression Program ver. 4.9.0.0 [22]. The linear trend was analyzed using Poisson regression and found to be significant. For this analysis, the independent variable was the year, and IR (per 100,000) was the dependent variable. The evaluation years with zero IR records per region (province) were replaced by a value of 0.5 for each count, such that the analysis could be performed in the linear regression model (on a logarithmic scale). The annual percentage change (APC) and its respective 95% confidence intervals were calculated to describe the linear trends by period. However, when the confidence interval was zero, the null hypothesis was not rejected (APC = 0). Subsequently, the annual percentage change trend was classified as increasing (positive APC), decreasing (negative APC), or stationary. *p* < 0.05 was considered statistically significant.

### 2.4. Spatiotemporal Analysis

A retrospective spatiotemporal analysis was performed using SaTScan ver. 9.6 [23]. A discrete Poisson’s model was used to detect areas at high risk for ASF (cluster) occurrence over 21 years (2000–2020), with no geographic overlap. For this purpose, a province was considered a cluster analysis unit. The space–time scan statistical analysis projects a cylindrical window with a circular geographic base (which reflects the space scan) and height (which reflects the period) [24]. For each location and size of the scan window, the alternative hypothesis was an elevated risk inside the window compared with the region outside the cluster. The most likely cluster (primary cluster) is the window with the maximum probability, that is, the cluster with the lowest probability of being formed by chance. In addition to the most likely clusters, secondary clusters were identified and ordered based on likelihood ratio test statistics. The statistical significance of the clusters was evaluated using Monte Carlo simulations with 999 repetitions. The relative risk (RR) was determined for each generated cluster using the ratio of the estimated risk within and outside the clusters. RR analysis was performed using the formula described by Liu et al. (2018) [25]:$$RR=\frac{c/E\left(c\right)}{\left(C-c\right)/\left(E\left[C\right]-E\left[c\right]\right)\;}$$
where “*c*” is the number of cases observed within the cluster and “*C*” is the total number of cases in the dataset, and “*E*[*c*] = *C*”, because the analysis is conditioned to the total number of cases. Geographic spatial autocorrelation analysis was performed using the QGIS software ver. 3.12 (Open Source Geospatial Foundation, [OSGeo], Chicago, IL, USA).

## 3. Results

### 3.1. Spatial Distribution of ASF

The pig distribution census of the provinces of Mozambique, analyzed over 21 years (2000–2020), demonstrated the existence of pig herds in all provinces of the country, with a predominance in Tete, Zambézia, and Inhambane (Figure 2A). During this period, 28,624 ASF cases were reported and distributed across three regions of the country (Table 1). Among these regions, 64.9 (18,584/28,624), 17.8 (5085/28,624), and 17.3% (4955/28,624) of total cases were observed in the northern, central, and southern regions, respectively.

Among the 11 provinces in Mozambique, all positive ASF cases were recorded during the analysis period (Figure 2B). The provinces with the highest percentage of reported cases were Cabo Delgado (40.8% [11,667/28,624]), Nampula (24.1% [6897/28,624]), and Maputo (11.7% [3357/28,624]) (Table 1). Niassa province had the lowest frequency of case notifications (20 cases), followed by Inhambane (122 cases) and Zambezia (154 cases). The IR analysis of the provinces showed a heterogeneous distribution at the country level (Figure 2C). The provinces with the highest IR values were Cabo Delgado (17,301.1/100,000 pigs), Maputo (8868.6/100,000 pigs), and Nampula (4588.7/100,000 pigs) (Table 1). At least one province with the highest IR was located in each region: Cabo Delgado and Nampula in the north, Manica in the central region, and Maputo in the south.

### 3.2. ASF Temporal Analysis

For the 21-year study, ASF cases were reported every year, except in 2000, when no cases were reported in any of the provinces of Mozambique (Figure 3). In 2020, Maputo reported an IR of 190 cases per 100,000 pigs. However, due to the small size of the city, the representation may not be clearly visualized (Figure 3). When analyzing the spatial distribution by region, the north recorded the first occurrence of ASF in Nampula in 2002 and Cabo Delgado in 2003 and 2004. Notably, Niassa had an IR of 101.3/100,000 in 2012 with a null risk in the remaining years. In contrast, during the period analyzed, cases of ASF were observed in Nampula in alternate years.

In the central region, the Zambézia and Sofala provinces were the first to report ASF cases from 2001 onwards. Subsequently, Zambezia was marked by alternations in the occurrence of IR, highlighting the five years (2001, 2004, 2011, 2015, and 2017). The provinces of Manica and Sofala were characterized by variations in the amplitude of the IR and reached higher values of the risk range throughout the central region. Together, these two provinces had the highest IR in 2006 and 2013. In contrast, Tete province showed an increase in IR between 2003 and 2005, with no subsequent cases until 2010.

In the south, the provinces of Maputo and Maputo city had the highest IR in the region for 16 years. Over seven years, Maputo province had the highest IR amplitude in the southern region (Figure 3). The Gaza and Inhambane provinces showed wide variations in positive and negative ASF cases. During the period evaluated (2000–2020), there were years with ASF-positive animals but also years without cases of infection in these regions.

Overall, during the analysis period (2000–2020), the distribution of ASF in the provinces of Mozambique was heterogeneous. However, a connection between neighboring provinces and the number of positive cases was observed in 2006, 2011, 2016, and 2017. Among them, positive cases in border provinces that connected the northern, central, and southern regions, forming a “corridor of positives”, were only observed in 2011. However, the highest number of affected provinces (seven) was observed in 2017 in addition to 2011.

Upon evaluating the IR of the provinces in relation to time (years), Cabo Delgado had the highest IR (8197.67/100,000 pigs) in 2004 (Figure 4). In the northern region, Nampula had the second-highest IR (3887.90/100,000 pigs). From the 16 years of data showing the spatial occurrence of ASF (2004–2019), it appears that positive cases appeared in consecutive years, except in 2012, with an IR of 2082.74/100,000 pigs in the Maputo province. Finally, throughout the period, Niassa province presented the lowest IR (100.57/100,000 pigs) in 2012.

### 3.3. Spatiotemporal Analysis of ASF

Spatiotemporal analysis revealed the formation of three clusters of high-risk ASF occurrences between 2000 and 2020, distributed in the northern, central, and southern regions of Mozambique (Figure 5). The primary cluster (Figure 5A) occurred in the northern region, encompassing the Cabo Delegado and Nampula provinces. This cluster revealed an RR of ASF infections that was 55.95 times higher in 2002 and 2014 compared with regions outside the cluster for the same years. The secondary cluster (Figure 5B) occurred in the southern region of the country in Maputo province and Maputo city, revealing an RR that was 6.04 times higher between 2004 and 2013.

Another secondary cluster was formed in the central region of the country, comprising the provinces of Manica and Sofala (Figure 5C), with an RR for ASF infection 7.58 times higher for pigs within the cluster in 2006.

### 3.4. Analysis of Temporal Trends

An analysis of the temporal trends in the regions and provinces of Mozambique between 2000 and 2020 revealed two trends, decreasing and stationary (Table 2). with the exception of Sofala (APC: –1.2%), Inhambane (APC: 9.4%), and Maputo (APC: 1.9%), which showed a stationary trend, the provinces showed a decreasing trend in annual IR during the evaluated period.

During the temporal evolution of IR, all provinces in the northern region showed a decreasing tendency in the number of ASF cases during the 21 years. In contrast, the central region showed a stationary IR trend only in Sofala, whereas the provinces of Inhambane and Maputo showed a static trend in the southern region.

## 4. Discussion

African swine fever is one of the most important diseases limiting swine production across the African continent [3,26] due to its recurrence and potential to spread across borders [1,27]. Europe has been responsible for 67% of outbreaks in the past five years. However, the most significant impact of ASF in terms of economic losses was reported in Asia (6,733,791 animals lost), representing 82% of total losses worldwide [27]. This implies that the disease has a global distribution pattern with severe repercussions on the swine industry worldwide. Due to its extensive infectivity [26,27,28], it can spread across different continents. The spatial characterization of ASF in affected countries allows the optimization of geographic surveillance measures to prevent its spread. In this study, the Mozambican provinces with the highest number of reported cases were Cabo Delgado (40.8%), Nampula (24.1), and Maputo (11.7%) (Figure 2B). Notably, the highest number of cases did not occur in the provinces with the highest cumulative numbers of swine, particularly Tete, Zambezia, and Inhambane (Figure 2A). Among these provinces, Tete and Zambezia border the Republic of Malawi, a country that reports endemic diseases and a sylvatic cycle between wild boars and soft ticks [3,29]. This finding implies the potential maintenance of the ASFV in wild species that circulate across borders with Mozambique, thereby increasing the risk of infection in domestic herds in these provinces. According to Schwarz et al. (2021), transboundary diseases, including ASF, are of great spatial importance due to their ability to spread infections through contact between susceptible individuals or the consumption of their by-products [30].

In the analysis of the total IR for ASF over the 21-year study (Figure 2C), the provinces with the highest risk of disease occurrence were spatially connected across borders, highlighting the following areas: (i) Cabo Delgado and Nampula, (ii) Sofala and Manica, and (iii) Maputo and Maputo city. This shows that ASFV is distributed in a potentially interconnected manner between regions and provinces and demonstrates the ability to spread among herds from different areas. Thus, although the northern region had the highest IR, the central region and the southern regions also had provinces with high IR, suggesting the need for an increase in targeted control and prevention measures.

Although ASF has been reported throughout the national territory, comprising the three regions of Mozambique, the data are underreported as most regions include small farmers where ASF-related swine deaths go unreported, allowing the proliferation and maintenance of the virus [3]. This has led to outbreaks with high economic losses, for example, in Cabo Delgado province, which reported outbreaks in five years (2003, 2004, 2007, 2008, and 2012) but represents 64.9% (18,584/28,624) of the total reported cases in the country.

In this study, the central and southern regions reported ASF occurrence for 16 years in at least one province (Figure 3). In the central region, ASF maintenance is related to the sylvatic cycle in populations of wild boars. In Gorongosa National Park, in Sofala province, Quembo et al. (2017) detected an ASFV seroprevalence of 9.1% in domestic swine and 78% in wild boar, demonstrating that the permanence of the sylvatic cycle is an important source for the maintenance of the disease in domestic swine [11]. In contrast, in the southern region, the Maputo province had a high frequency of case reports from 2004 to 2019. A study described that an ASF outbreak re-emerged in 2004 in a village in Maputo city, spreading in subsequent years to the Maputo province [3].

Notably, no ASF cases were reported in the province in 2020. However, an IR of 190 cases per 100,000 pigs has been reported, demonstrating the difficulty in controlling ASF in the region. In the southern region, the source of infection for domestic pigs seems to be the movement of infected pigs from rural areas raised in traditional systems, which allows the free movement of animals between herds of different origins, leading to a higher risk of viral transmission between farms [31,32].

The ASF spread data analysis conducted in this study is more relevant, as three viral genotypes were confirmed in a genetic characterization analysis: (i) genotype II p72, (ii) p72 V genotype, and (iii) genotype XXIV p72. Quembo et al. (2017) identified genotype XXIV p72 as being responsible for outbreaks in Mozambique (strain MOZ 2/2002) [11] since 2002 and over the last two decades in Madagascar (strain MAD/1/98) and Mauritius (strain MAU1/2007) [33]. This approach reinforces the need for cross-border disease control to prevent the viral strains in Mozambique from being transmitted to other African countries.

Although ASF has been reported in almost all provinces of Mozambique, Niassa reported only a few cases (20 cases) in the 21 years analyzed. These data should be evaluated with caution, as a low notification rate does not necessarily imply that there were no cases. According to MAPA (2009), these regions are called “silent areas,” where there are no reported cases of the disease, but the pathogen spreads spatially without proper notification, causing outbreaks [34,35,36]. Similarly, provinces with a high number of reported cases may imply the absence of effective measures to control the disease. However, this may also be related to the greater efficiency of official surveillance. In Maputo province, where family swine farms are represented mainly by peri-urban swine farms, animal healthcare professionals likely have greater control over ASF cases, even if attempts are made to hide them, due to the understanding that animals positive for ASF should be slaughtered [3].

The spatiotemporal analysis identified three clusters with a high risk for the occurrence of ASF between the periods evaluated, distributed in three regions: northern, central, and southern. The primary cluster, formed at the beginning of the analyzed period (2002–2014), covered the provinces of Cabo Delgado and Nampula (Figure 5A), with an RR 55.9 times higher than that of the other provinces of the country. These data alerted Tanzania to the ASF risk along the northern border of Mozambique. Despite Cabo Delgado having the lowest average swine population (46,536.2) followed by Nampula (116,046.4), it showed a high RR for a long time (13 years). A higher animal population density is crucial, but not deterministic, for increasing the number of infections in an animal population because well-established biosecurity measures can control the incidence of cases [37].

Subsequently, a secondary cluster covering Maputo province and Maputo city (Figure 5B) was formed between 2004 and 2013, with an RR of 6.04. The secondary cluster (2004–2013) coincided with some years of the primary cluster (2002–2014), demonstrating that both the northern and southern regions had a long RR period for pigs to be infected. The disease is much more likely to be reported in commercial farms than in family farms, as these have a high economic value ratio and are usually located close to cities where veterinary services are well-represented, enabling owners to seek veterinary care when their animals show clinical signs as seen in Maputo province and Maputo city. This may explain the higher frequency of ASF reports in the region of the formed cluster than in more remote areas where veterinary services are poorly represented [11].

Another secondary cluster formed in the central region in 2006 in the provinces of Manica and Sofala, with an RR of 7.58. The cluster formation area proved to be strategic for Mozambique to control pig intercommunication in both extreme regions: north and south. This is because the Central region is a corridor for the movement of animals and their contaminated by-products from the north to south and vice versa. In addition, there was border in province of Tete, with frequent reports of ASF [3]. Currently, animal health education strategies have been implemented in the northern region of Tete Province in the district of Angónia to reduce the risk of transmission of infectious agents, such as the ASFV, to herds of small swine producers [38].

Wild boar populations are higher in many African countries. The National Park of Gorongosa in the Sofala province houses a large number of these suid species. The transmission of ASF between boars and domestic pigs is significant due to the lack of biosecurity in smallholder pig farming; therefore, any positive wild animal can spread the disease and cause problems in the region. Thus, this scenario allows for the interaction between wild boar and domestic pigs, which can be responsible for the transmission of ASFV between the wild and domestic suids [39]

In Mozambique, the establishment of trade routes for pigs between districts, from the interior to cities or outside the province, is common. Traders often buy live pigs from various suppliers along these routes and transport them to their respective cities for sale in informal markets. Pigs that die along the way are sold during their journey or upon arrival in the city [3]. Unfortunately, the sale of pigs with clinical signs of ASF may be the reason for the higher occurrence of ASF in the region and, consequently, its spatial dissemination.

Temporal trend analysis revealed that most provinces in Mozambique have been experiencing a downward trend in ASF cases over the last two decades. However, the trends in the Sofala, Inhambane, and Maputo provinces tend to be stationary. This must be carefully analyzed as it could imply that the disease is being controlled or that it has stabilized at a high frequency. This trend was verified in Sofala and Maputo, where the stability of cases had high IRs of 1691.0/100,000 pigs and 8868.6/100,000 pigs, respectively. The trend toward the high endemicity of ASF in these regions indicates more effective government control to reduce the occurrence and maintenance of cases in the region.

Nearly all regions of Mozambique have movement of animals and their by-products as one of the main risk factors for the transmission and spread of ASF. This is particularly observed in the traditional culture of almost all producers in the family sector [12,31,32]. Therefore, it is likely to be a key factor in disease occurrence in areas of viral circulation throughout the country. Decree 26/2009 of the Animal Health Regulations of the Mozambican Government established measures for sanitary surveillance and disease control in cases of outbreaks. According to the decree, prompt diagnosis, slaughter, compensation, and disposal of all animals in infected facilities must be offered. The legislation additionally mentions the complete cleaning and disinfection of farms, restriction of movement, and surveillance. These measures constituted attempts by the government to control the disease in areas of high viral transmission. However, the lack of a strategy on the part of the government to coordinate actions, such as a weak logistical policy for compensation to affected breeders and a deficient network of national veterinary assistance for disease intervention, should be considered as probable reasons for the increased risk of ASF in each region, as indicated by the findings of this study.

The cultural, socio-economic, and environmental characteristics of pig farming must be considered in Mozambique. Although the analyzed data were obtained from the official database of the Mozambican government, the aggregation of ASF cases by province did not make it possible to assess the risks and trends at the district level or the specific risk factors of each swine herd with positive cases. Schwarz et al. (2021) [30] and Oliveira et al. (2022) [37] also noted this limitation when analyzing data from official secondary data sources in Brazil using the same methodology. Although official ASF notifications by the Ministry of Agriculture are essential for evaluating the performance of the control program in the country, the underreporting of cases must be considered, especially when there is no accountability to the owners of the affected animals. The results of this study make it possible to evaluate ASF in swine, implement geographic surveillance, and evaluate its spatial distribution and risks to control the spread of the disease across borders. Furthermore, federal programs to prevent and control ASFV infections could be the next step in raising awareness and preparedness for farmers, veterinarians, and farm workers in Mozambique. The actions for ASF control suggest the effective allocation of financial resources, alerting government agencies about possible ASF risks in animal health control programs.

## 5. Conclusions

In Mozambique, ASF is spatially heterogeneously distributed, encompassing regions at high risk of infection in the northern, central, and southern regions, suggesting a threat to the swine production sector in all regions. Historically, ASF has been a significant problem for the country since its control is limited by problems of health education, capillarity of swine movement, and because wild boars maintain the sylvatic cycle in domestic swine. The spatial and temporal evaluation of ASF in Mozambique also reveals that funds can be better applied when directed to regions with a high risk of infection, as well as delimiting strategies to control the spread of the agent beyond the borders of the country. Therefore, it must be emphasized that the ASF control measures should not only be the exclusive responsibility of the veterinary services, but the whole swine production chain must take precautions to ensure that their herds are not infected. In addition, this study suggests implementing more effective actions for epidemiological investigation and notification of ASF in provinces with low reporting frequency through financial incentives, professional training, and health education campaigns for producers and rural populations. To the best of our knowledge, this is the first study to evaluate the spatial distribution of ASF in different regions of Mozambique at the provincial level.

## Figures and Tables

**Figure 1 viruses-15-01010-f001:**
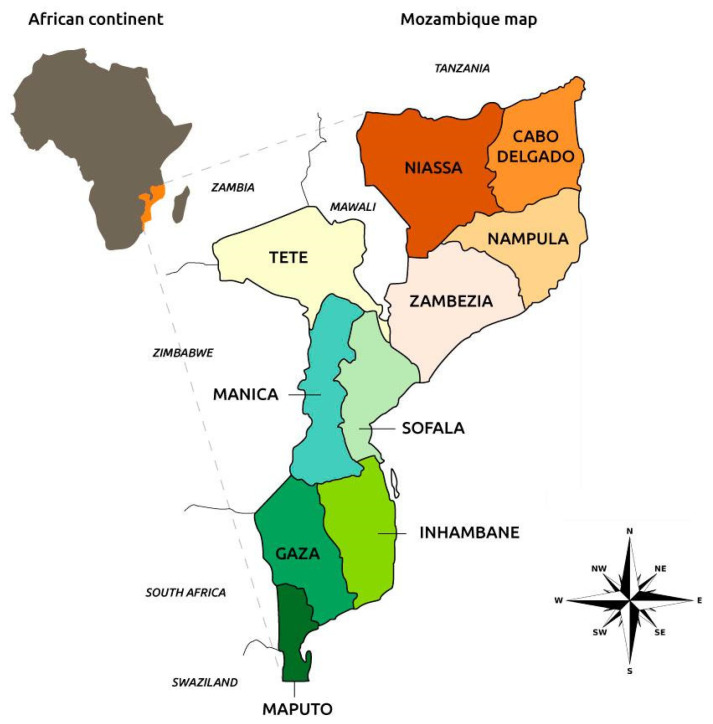
Geographic boundaries and provinces of Mozambique.

**Figure 2 viruses-15-01010-f002:**
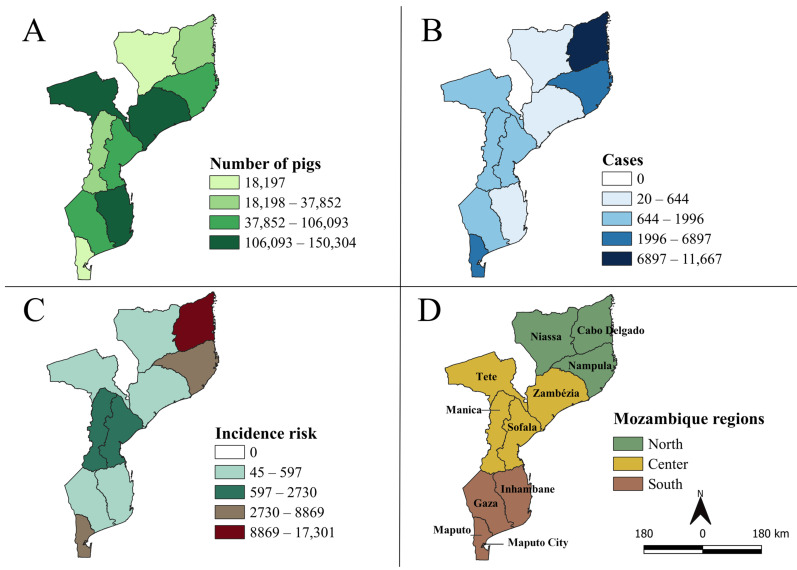
Spatial distribution of the average swine population (**A**); the total number of accumulated African swine fever (ASF) cases (**B**); incidence risk of ASF per 100,000 pigs (**C**); political division of Mozambique (**D**) between 2000 and 2020.

**Figure 3 viruses-15-01010-f003:**
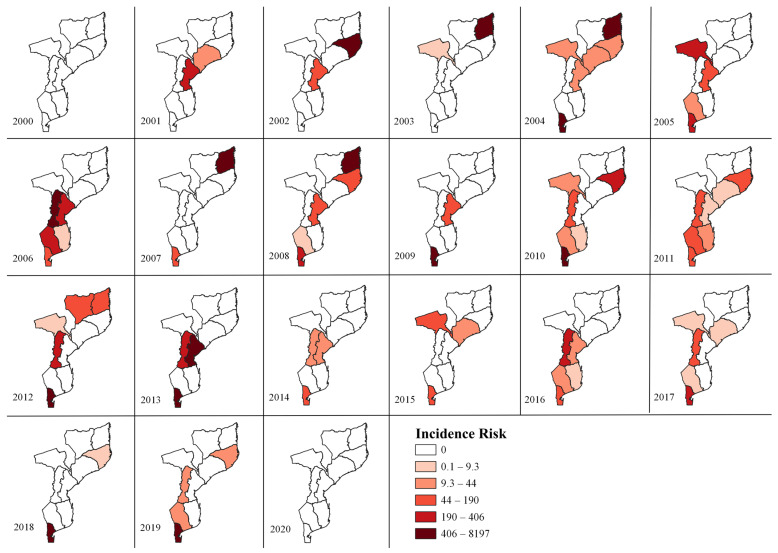
Spatial distribution of incidence risk (IR) of African swine fever per 100,000 pigs between the years 2000 to 2020 in the provinces of Mozambique. Note: In 2020, the IR was 190.78 in Maputo City. However, due to the size of the regions associated with the scale of the map, it is not possible to visualize the color scale.

**Figure 4 viruses-15-01010-f004:**
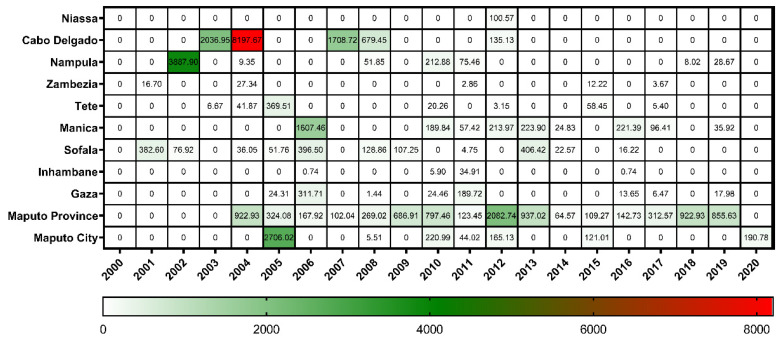
Total incidence risk (IR/100,000 animals) of African swine fever in pig population according to the analysis year and Mozambique provinces between 2000 and 2020.

**Figure 5 viruses-15-01010-f005:**
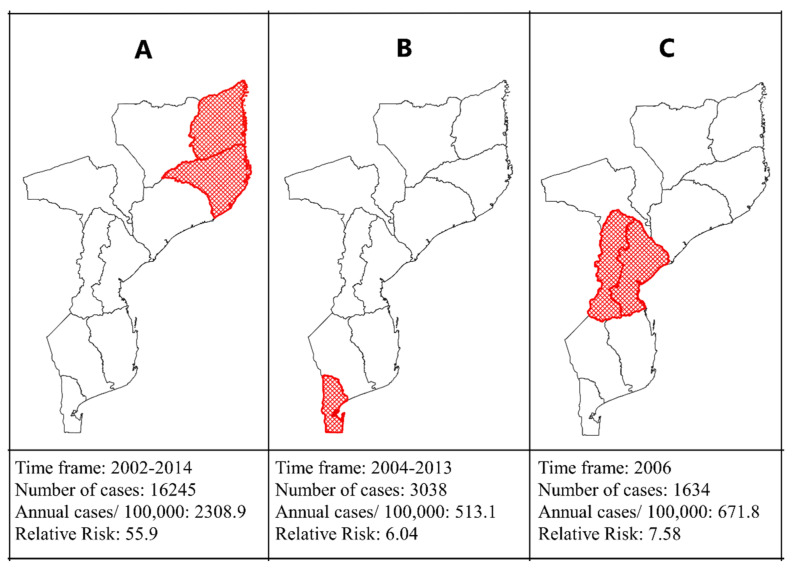
Spatiotemporal distribution of African swine fever in Mozambique between 2000 and 2020. Red represents clusters formed: primary (**A**) and secondary clusters (**B**,**C**).

**Table 1 viruses-15-01010-t001:** The total number of cases, relative frequency (RF), and incidence risk (IR) of African swine fever in pigs in different regions and provinces of Mozambique between 2000 and 2020.

Regions/Provinces	Cases	RF (%)	IR (per 100,000)
North	18,584	64.9	7825.4
Niassa	20	0.1	101.3
Cabo Delgado	11,667	40.8	17,301.1
Nampula	6897	24.1	4588.7
Central	5085	17.8	786.4
Zambézia	154	0.5	62.9
Tete	1141	4.0	513
Manica	1996	7.0	2730.4
Sofala	1794	6.3	1691
South	4955	17.3	1066.9
Inhambane	122	0.4	45.4
Gaza	832	2.9	596.7
Maputo Province	3357	11.7	8868.6
Maputo City	644	2.2	3539

**Table 2 viruses-15-01010-t002:** Time trend analysis of African swine fever using Joinpoint regression in Mozambique, 2000–2020.

Province	APC	Lower CI	Upper CI	*p*-Value	Trend
Niassa	−22.3 *	−28.3	−15.8	<0.001	Decreasing
Cabo Delgado	−31.2 *	−43.7	−16.1	0.001	Decreasing
Nampula	−32.1 *	−37.4	−26.4	<0.001	Decreasing
Zambézia	−9.5 *	−17.4	−0.7	0.035	Decreasing
Tete	−17.7 *	−28.9	−4.6	0.012	Decreasing
Manica	−23.2 *	−28.8	−17.3	<0.001	Decreasing
Sofala	−1.2	−9.9	8.3	0.780	Stationary
Inhambane	9.4	−17.2	44.4	0.508	Stationary
Gaza	−15.9 *	−26.8	−3.3	0.018	Decreasing
Maputo Province	1.9	−6.2	10.6	0.647	Stationary
Maputo City	−22.3 *	−28.3	−15.8	<0.001	Decreasing

* Significant trend (95% CI).

## Data Availability

Not applicable.

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
