# Peer review of "High-Risk Regions of African Swine Fever Infection in Mozambique"

_viruses, 2023, doi:10.3390/v15041010_

Round 1
Reviewer 1 Report (New Reviewer)
Dear authors,
thank you very much for this interesting manuscript.
I would like to submit to your attention some suggestions, as following.
Case: I suggest to clearly define primary and secondary cases.
Awareness and Preparedness: I suggest to stress this point, because it is of paramount importance for ASF prevention and control.
Role of wild boar: I suggest to better describe why wb are considered a risk factor for Dp
Minor revisions:
· LINE 65: “...it is suspected that the virus is maintained by the sylvatic cycle of domestic swine...”maybe you wanted to say “in domestic swine?”
· TABLE 1: Cases for Sofala: 1 is missing (“.794”)
Author Response
Dear Reviewer,
Thank you for carefully reading our manuscript. Many thanks for the suggestion. Below, there are the answers to your questions.
Question 1
Case: I suggest to clearly define primary and secondary cases.
Answer: Many thanks for the suggestion. Due to the characteristic of high propagation of the African Swine Research virus, the official data obtained by the government of Mozambique does not distinguish between primary and secondary cases. Therefore, it is not possible to identify which was the primary (original) case in each region. However, for the purpose of our study, regardless of whether the case is primary or secondary, all focus control measures are the same.
Question 2
Awareness and Preparedness: I suggest to stress this point, because it is of paramount importance for ASF prevention and control.
Answer: Thank you. We wrote the last paragraph of discussion about your suggestion. You may see in the manuscript.
“ Furthermore, federal programs to prevent and control ASFV infections could be the next plan to raise awareness and preparedness for farmers, veterinarians and farmworkers in Mozambique. The actions for ASF control suggest the effective allocation of financial resources and alerts government agencies about possible ASF risks in animal health control programs.”
Question 3
Role of wild boar: I suggest to better describe why wb are considered a risk factor for Dp
Answer: Thank you. Done. You may see in the manuscript.
“It is estimated that the wild boar population is higher in many countries in Africa. The National Park of Gorongosa in Sofala’s province presents a high number of this suid species. Because the transmission of ASF between boar and domestic pig is underestimated, any positive wild animal can spread the disease and cause several problems in the region due to the lack of biosecurity in smallholder pig farming. Thus, this scenario allows interaction between wild boar and domestic pigs and can be responsible for the connection between the wild and domestic cycles [39].”
Question 4
LINE 65: “...it is suspected that the virus is maintained by the sylvatic cycle of domestic swine...”maybe you wanted to say “in domestic swine?”
Answer: Thank you. Done. You may check the manuscript.
Question 5
TABLE 1: Cases for Sofala: 1 is missing (“.794”)
Answer:: The typo in Table 1 has been corrected. Thank you very much.
Answer:: The typo in Table 1 has been corrected. Thank you very much.

Reviewer 2 Report (New Reviewer)
Aritcle
“High-Risk Regions of African Swine Fever Infection in Mozambique” by Mataca et al.
The presented project falls well within the scope of Viruses.
Due to the spread of African Swine Fever in wild and domestic swine with tremendous socio-economic losses, it’s important to know the regions with a high risk of outbreaks or introductions. Therefore, the study evaluated the spatiotemporal distribution and temporal trends of ASF in swine in Mozambique between 2000 and 2020.
General scientific value
The topic is of great interest and the results provide an advancement of the current knowledge. The methods are described well.
Major Points
In general, it would be very interesting to know in which region, which ASFV genotype occurred and if there is an influence in the IR, RF and Spatiotemporal analysis depending on the genotype. The publication would be greatly enhanced if these data could be included.
Minor Comments
Line 50: please change and from italic form to normal
Table 1: fist line IR ( /100,000) please add cases
Table 1: please check the number of cases at Sofala
Table 1: please change Maputo Provincia to Maputo Province and Maputo cidade to Maputo city (Also in fig 4)
Line 246: Please check the dates, because according to fig. 3 also the years 2004,2006, 2008, 2012, 2016, 2017 show 5 affected provinces and 2010 there were 6 provinces affected. Please clarify.
Line 266, Line 268 and Fig 5: Please check the time frame if it’s correct
Author Response
Comments to the author from reviewer 2
We appreciate your comment and agree with you. Your critic was very helpful. We have done major alterations in the results and discussion section. The alterations are highlighted in yellow.
Question 1
In general, it would be very interesting to know in which region, which ASFV genotype occurred and if there is an influence in the IR, RF and Spatiotemporal analysis depending on the genotype. The publication would be greatly enhanced if these data could be included.
Answer: You are right. We agree with you. However, the goal of our work is to calculate classical epidemiological information. We hope to call the attention of the Ministry of Agriculture and other international agencies for ASF control in Mozambique. After them, It could be possible to develop further strategies to try to resolve the problem and contain the virus. Is interesting the control of ASFV in Mozambique for other parts of the world. The infection could be a transboundary disease for other countries in the rest of the world from Mozambique. Furthermore, the results of this paper could help us get grants (budget) to develop studies of genetics of ASFV genotype and molecular epidemiology in Mozambique. So, NGS of viral RNA could be the next step of the project. Another point, the Ministry of Agriculture in Mozambique did not keep the ASFV samples. We don't have access to these samples. The work about the RNA viral genome must be prospective with new samples.
Question 2
Minor Comments
Line 50: please change and from italic form to normal
Answer: Many papers write genus and family for viruses in normal letters. If you check in different papers, you will probably see them in regular letters. However, we are following the International Committee on Taxonomy of Viruses (ICTV) criteria that asks us to write in italic format. You may check the roles in the website (https://ictv.global/taxonomy). The virologist writes in italic format. Please let me know if you agree with us.
Question 3
Table 1: fist line IR ( /100,000) please add cases
Answer: In Table 1, (/100,000) refers to a proportional unit for each of the values in the column (IR). Thus, each value in the column is proportionally related to 100,000 animals. Thus, there is no way to put only a number of cases. Please let me know if you agree with the presentation of data. Otherwise, I can repeat the proportional unit (/100,000) in each line. Thank you very much.
Question 4
Table 1: please check the number of cases at Sofala
Answer: Thank you very much. You are right. The typo in Table 1 has been corrected.
Question 5
Table 1: please change Maputo Provincia to Maputo Province and Maputo cidade to Maputo city (Also in fig 4)
Answer: Thank you very much. You are right. We did the correction. You may check the manuscript.
Question 6
Line 246: Please check the dates, because according to fig. 3 also the years 2004,2006, 2008, 2012, 2016, 2017 show 5 affected provinces and 2010 there were 6 provinces affected. Please clarify.
Answer: Thanks for your questions. The dates are correct. However, the sentence was probably a little confusing. We rewrite the sentence to improve the understanding. Please, check the new sentence.
Question 7
Line 266, Line 268 and Fig 5: Please check the time frame if it’s correct
Answer: The time frame is correct. You can see in Fig. 5 that the province of Gaza and Inhambane was unstable for cases of ASF during the evaluation period (2000-2020). There were years with positive cases and also years with just negative cases. We wrote the sentence to better understand. Let me know if you agree with the new sentence.
“The provinces of Gaza and Inhambane were unstable by variations in the positive cases and negative cases. During the period evaluated (2000-2020), there were years with positive animals, but also years without cases of infection in different regions.”
Question 8
(x) English language and style are fine/minor spell check required
Answer: We sent the manuscript to the Editage company. The certification is attached.

Reviewer 3 Report (New Reviewer)
It is clear that African swine fever (ASF) is a serious infectious disease for countries and especially in countries with a great shortage of food countries.
In my opinion the proposed study has highlighted a series of circumstances that will certainly be useful for the development of further strategies to try to resolve the problem and for developing a strategy to contain the virus.
The percentages shown in the analysis of the methods described concerning African swine fever (ASF), accurately describe the existing scenario and will certainly be a guideline for future studies
Author Response
Comments to the author from reviewer 3
It is clear that African swine fever (ASF) is a serious infectious disease for countries and especially in countries with a great shortage of food countries.
In my opinion the proposed study has highlighted a series of circumstances that will certainly be useful for the development of further strategies to try to resolve the problem and for developing a strategy to contain the virus.
The percentages shown in the analysis of the methods described concerning African swine fever (ASF), accurately describe the existing scenario and will certainly be a guideline for future studies
Dear Reviewer,
Thank you to help us to improve the quality of manuscript. We wrote some parts of manuscript. In addition, We sent the manuscript to the especialized company in english correction.

Round 2
Reviewer 2 Report (New Reviewer)
Dear Authors,
Thank you very much for your answers and corrections!
Q1: I am sure it was just a typo, however if you want to include sequencing in future studies, which would be of very interest, please use the DNA because it’s a ds DNA virus and not a RNA virus.
Q2: The reviewer agrees with the authors that genus and family is written in italic, that´s correct. I only meant the word “and”. Anyway, you changed the sentence, so its fine.
Q3-8: The other improvements are fine for the reviewer.
This manuscript is a resubmission of an earlier submission. The following is a list of the peer review reports and author responses from that submission.
Round 1
Reviewer 1 Report
The authorsuses an analysis of data on the occurrence of African swine fever in various regions within Mozambique over a period of time, calculates the risk of incidence, performs a spatio-temporal analysis and compares the situation in different regions. The overall structure and content of the paper is sound.
Through the article I believe that the raw data are differentiated by administrative divisions at the provincial level, but in animal epidemiology studies this is usually considered to be a rather crude analysis. It is recommended that the authors further study the prevalence of African swine fever in different areas of the provinces with higher risk and use spatial analysis to find the centre of local prevalence, which would more accurately reflect the risk of African swine fever in a specific spatial and temporal context.
The authors mention some of the animal health measures taken in Mozambique in response to African swine fever, but the content is not sufficient. It is recommended that the discussion section should include information on animal disease surveillance and control measures in each region, and that analysis and evaluation should be carried out to explore the effectiveness of the different measures implemented and their impact on the prevalence of African swine fever.
Author Response
Dear reviewer 1, thank you for helping us improve the quality of our manuscript.
Comments to the author from reviewer 1.
Question 1
Through the article I believe that the raw data are differentiated by administrative divisions at the provincial level, but in animal epidemiology studies this is usually considered to be a rather crude analysis. It is recommended that the authors further study the prevalence of African swine fever in different areas of the provinces with higher risk and use spatial analysis to find the centre of local prevalence, which would more accurately reflect the risk of African swine fever in a specific spatial and temporal context.
Answer: Thank you for carefully reading our manuscript. Many thanks for the suggestion. It is a great question. However, in Mozambique, positive cases are obligatorily reported to the Official Veterinary Service, and control measures are applied in accordance with the parameters recognized by the OIE. Thus, each case represents a new case, and every time this data enters the official system (which was used in the present study), the animal has already been sacrificed. Therefore, epidemiologically, the incidence is equal to the prevalence, as positive animals are not allowed to be kept on properties. The applied methodology is widely used in animal health defense programs since it uses reported official data and does not actively search, mainly in complex diseases such as ASF that use several hosts: domestic, wild, and vectors that can harbor the virus and alter the immediate prevalence. Thus, our study aimed to direct the risks of ASF infection in relation to the different regions, and the use of incidence is more accurate when compared to the prevalence for this methodology to ASF. Furthermore, the use of incidence as a reference parameter for regions at high risk of infection has already been extensively used in the scientific literature for the purpose of this study, such as:
1 - Fonseca-Rodríguez, O., Pinheiro Júnior, J.W., Mota, R.A., 2019. Spatiotemporal analysis of glanders in Brazil. J. Equine Vet. Sci. 78, 14–19.
https://doi.org/10.1016/j.jevs.2019.03.216.
2 - Silva, Ana Elisa Pereira da, Chiaravalloti Neto, Francisco Conceição, Gleice Margarete de Souza. Leptospirosis and its spatial and temporal relations with natural disasters in six municipalities of Santa Catarina, Brazil, from 2000 to 2016. Geospatial Health, v. 15, n. 2, p. 225-235, 2020. https://doi.org/10.4081/gh.2020.903.
3 - Schwarz, D.G.G., Sousa Júnior, P.F., Silva, L.S., Polveiro, R.C., Oliveira, J.F., Faria, M.P. O., Marinho, G.L.O.C., Oliveira, R.P., Moreira, M.A.S. 2021. Spatiotemporal distribution and temporal trends of brucellosis and tuberculosis in water buffalo (Bubalus bubalis) in Brazil. Prev. Vet. Med. 193, 105417.
https://doi.org/10.1016/j.prevetmed.2021.105417.
4 - Pei, X.; Li, M.; Hu, J.; Zhang, J.; Jin, Z. Analysis of Spatiotemporal Transmission Characteristics of African Swine Fever (ASF) in Mainland China. Mathematics 2022, 10, 4709.
https://doi.org/10.3390/math10244709
Question 2
The authors mention some of the animal health measures taken in Mozambique in response to African swine fever, but the content is not sufficient. It is recommended that the discussion section should include information on animal disease surveillance and control measures in each region and that analysis and evaluation should be carried out to explore the effectiveness of the different measures implemented and their impact on the prevalence of African swine fever.
Answer: We modified it as you can see in manuscript at the discussion section.
You may check the modifications in file attached.

Reviewer 2 Report
1. The introduction in "2.1. Study area" does not need to present the farming situation of other animals, but should be more specific and complementary to the data on pig farming in each province in each year.
2. The original data in "2.2. Data source" needs to be uploaded to the supplementary file.
3. The spatial distribution of African swine fever is related to many factors, such as climate, pig breeding density, pig prices and other factors, so purely analysis from apparent data does not have reference value.
Author Response
Dear Reviewer 2, thank you for considering our manuscript and help us with quality
Question 1
The introduction in "2.1. Study area" does not need to present the farming situation of other animals, but should be more specific and complementary to the data on pig farming in each province in each year.
Answer: Thank you for your suggestion. We agreed with you and we modified the manuscript.
Question 2
The original data in "2.2. Data source" needs to be uploaded to the supplementary file.
Answer: It is a great idea. We attached the file in supplementary material.
Question 3
The spatial distribution of African swine fever is related to many factors, such as climate, pig breeding density, pig prices and other factors, so purely analysis from apparent data does not have reference value.
Answer: ASF is an important disease that has a large impact on economy of many countries in the world, especially in countries from Europe and Asia. In our work, we identified that Mozambique has regions with a high risk of virus shedding to borders. It is important information to rest of world. In attention, if different countries and authority wants ASF eradication in the world. We believe that ASF is an animal disease negligencied in AFRICA countries by rest of the world.

Round 2
Reviewer 1 Report
Thank you for the author's response. It is a pleasure to continue discussing these with you. In the field of African swine fever epidemiological research, the focus of analysis has usually been on the patterns of virus transmission and their influence by climate, the natural environment, vector organisms, and the management practices of pig production, transport of live animals and products, and people's consumption habits in a given region.
As a researcher in the field of animal epidemiology, you must be well aware that the prevalence of African swine fever under certain spatial and temporal conditions is influenced by a combination of many factors, and that many different types of data are required to study and analyse these factors. I think the main problem with this paper is that it needs to be analysed with more specific data. It is more important to discuss the reasons for the prevalence of African swine fever in certain areas after the results have been calculated using spatio-temporal analysis tools.
Unfortunately, You also mentioned, "In Mozambique, the cultural, socio-economic, and environmental characteristics of pig farming must be considered. Although the analyzed data were obtained from the official database of the Mozambican government, the aggregation of ASF cases by province did not make it possible to assess the risks and trends at the district level or the specific risk factors of each swine herd with positive cases". Therefore, the reader cannot obtain more innovative and inspiring content from this paper. I recommend that you should submit this paper to another journal.